# Light Weight, Flexible and Ultrathin PTFE@Ag and Ni@PVDF Composite Film for High-Efficient Electromagnetic Interference Shielding

**DOI:** 10.3390/ma16134831

**Published:** 2023-07-05

**Authors:** Hongbo Liu, Jiajie Huang, Bingzhi Guo

**Affiliations:** School of Materials and Environment, Beijing Institute of Technology, Zhuhai 519088, China; liuhongbojm@163.com (H.L.); 200501102634@bitzh.edu.cn (J.H.)

**Keywords:** electromagnetic shielding, Ag, PTFE, composite film

## Abstract

Dopamine was used to modify polytetrafluoroethylene (PTFE) in order to obtain functional polydopamine (PDA) surface-modified PTFE microporous film (PTFE@PDA). Ag was deposited on the surface of PTFE@PDA using electroless plating in order to obtain Ag-wrapped PTFE@PDA film (PTFE@Ag). A liquid-phase chemical reduction method was employed to prepare nickel nanochains. A Ni@PVDF cast film was obtained by mechanically blended nickel nanochains and polyimide (PVDF). The above two films were hot pressed to give a flexible, ultra-thin, and highly effective electromagnetic shielding composite film with a “3+2” layered structure. IR, XRD, and TEM results showed the PTFE@PDA film surface was coated by a tight plating layer of Ag particles with a particle size of 100~200 nm. PTFE@Ag+Ni@PVDF composite film exhibited excellent electromagnetic shielding effectiveness, with the conductivity of 7507.5 S/cm and the shielding effectiveness of 69.03 dB in the X-band range. After a 2000-cycle bending, this value still remained at 51.90 dB. Furthermore, the composite film presented excellent tensile strength of 62.1 MPa. It has great potential for applications in flexible and wearable intelligent devices.

## 1. Introduction

With the coming of the 5G era as well as the widespread use and rapid development of various electronic devices, electromagnetic radiation has become an increasingly prominent environmental concern. Long-term exposure to electromagnetic waves may lead to serious harm of human health, and electromagnetic wave leakage can also endanger information security [1]. In order to meet the ever-increasing requirements for electromagnetic compatibility in electronic communication products, the development of lightweight, flexible electromagnetic interference (EMI) shielding materials has attracted great attention of researchers [2,3,4]. It is believed that EMI shielding has three main mechanisms: reflection, absorption, and multiple reflections. According to the electromagnetic shielding mechanism, the shielding effectiveness (SE) heavily reflects the electromagnetic shielding performance of a material. Intrinsic conductivity of a material is generally considered to be one of the most critical parameters affecting SE, which is related to material surface structure and thickness [5]. It is well known that metals-based materials have excellent shielding performance because of their excellent conductivity, but problems such as high density, susceptibility to corrosion, and inadaptability to harsh environments make them limited for practical applications [6]. In recent years, conductive polymer composite films prepared from a polymer matrix and conductive fillers have received more and more attention due to their unique magnetic, mechanical, optical, and chemical properties [7,8,9,10]. At present, polyacrylonitrile (PAN) [9], polyvinylalcohol (PVA) [11], PTFE [12,13], PVDF [14,15], polybutylene terephthalate (PET) [16], etc., are used as polymer matrixes, and commonly used conductive fillers mainly include graphene [4,17,18], CNTs, MXenes [8,19,20,21,22,23], metals [16,24,25], alloys [14,26], perovskite metal oxides [27,28], and so forth. These conductive polymer composite films exhibit good EMI SE. For example, Tan H.X. et al. designed and prepared a sandwich-structured Ti_3_C_2_T_x_-MXene/reduced graphene oxide composite (MrG) membrane with high conductivities and abundant interior blowholes for high-performance EMI shielding and IR stealth. The membrane had distinctive porous structures and featured around an 18 μm thickness with a high EMI SE of 89.1 dB and a satisfactory absolute EMI SE (SSE/t) of up to 55,282.8 dB cm^2^/g. This was the best result for MXene-based nanocomposites with a similar thickness [19]. The distinctive porous structure was believed to play an initial role in improving the EMI SE.

PTFE has excellent characteristics such as high-temperature resistance, corrosion resistance, low adhesion, flow resistance, and friction coefficient [29,30]. PTFE-based electromagnetic shielding materials are believed to have great practical application potential [13]. However, they are prone to slippage between fillers and the polymer matrix under large deformation conditions, resulting in damage to the conductive network [31]. Therefore, electromagnetic shielding effectiveness decreases, and the metal layer can easily fall off, resulting in poor secondary processing. Studies have shown that dopamine would undergo an oxidation self-polymerization reaction to form a strong polydopamine layer (PDA) attached to the surface of solid materials under weak alkaline solution conditions [30,32]. The phenolic hydroxyl, amino, and imino functional groups in PDA could change the surface properties of the coated material, providing a chelating site with metal ions, as well as favorable conditions for subsequent modification operations such as grafting, plating, and coating [33].

Metal-filled conductive polymer composite film has good application prospects because of its low cost, wide frequency band, good performance, and non-polluting, lightweight properties. These properties make it possible to apply in different occasions and environments. As a filler material with excellent electrical conductivity, silver has been widely used in conductive films [23,24,25,34]. However, due to the high toxicity of stannous chloride in traditional silver-plating chemical reagents and the high price of palladium chloride, it is necessary to find more ideal alternatives in practical applications.

Considering the above issues, it is proposed to use dopamine to modify the PTFE film. Based on the hysteresis loss mechanism of nickel nanowires, silver nitrate was used as an activator to prepare the PTFE@Ag film via electroless plating. PTFE@Ag and Ni@PVDF film were hot pressed to produce a PTFE@Ag+Ni@PVDF composite film with excellent electromagnetic shielding effectiveness through a metal-wrapped polymer fiber structure. Our research team successfully synthesized PTFE@Cu+Ni@PVDF composite film via this method [35]. The prepared film has great conductivity (1117.57 S/cm) and shielding efficiency (57.16 dB) in the X-band range. It has a great potential application prospect for flexible devices and portable wearable intelligent devices. Therefore, the prepared PTFE@Ag+Ni@PVDF composite film is expected to obtain excellent electromagnetic shielding performance.

## 2. Materials and Methods

### 2.1. Materials

PTFE 0.1μm hydrophilic film (thickness 25~30 μm) was obtained from Asimo Technology Co., Ltd. (Shanghai, China). Dopamine hydrochloride (98%) was procured by Bomei Biotechnology Co., Ltd. (Hefei, China). Tris-HCl (99.5%) was obtained from Yami Biotechnology Co., Ltd. (Beijing, China). NaOH and absolute ethanol (AR) were acquired from Damao Chemical Reagent Factory (Tianjin, China). Ammonia (AR) and AgNO_3_ solution (0.5 mol/L) were supplied by Wengjiang Chemical Reagent Co., Ltd. (Shaoguan, China). Glucose (FG) was purchased from Weimeiyuan Food Ingredients Co., Ltd. (Shanghai, China). N-methylpyrrolidone (AR) was purchased from Yatai United Chemical Co., Ltd. (Wuxi, China). Ethylene glycol (AR) was obtained from Aladdin Biochemical Technology Co., Ltd. (Shanghai, China). Disodium ethylenediaminetetraacetic acid (EDTA) and NiCl_2_·6H_2_O (AR) were supplied by China National Pharmaceutical Group Chemical Reagent Co., Ltd. PVDF was purchased from Candlelight New Energy Technology Co., Ltd. (Dongguan, China).

### 2.2. Preparation of PTFE@PDA

PTFE hydrophilic film (4 cm × 4 cm) was soaked in absolute ethanol for 30 min in order to remove its surface oil. It was thoroughly washed with purified water under ultrasonic radiation, and dried in an oven at 45 °C. A total of 0.316 g of dopamine (DA) was dissolved with hydrochloride powder into 0.01 mol/L of Tris-HCl solution to form a transparent solution, and the pH value was kept to about 8.5 with 3 mol/L NaOH solution for the preparation of a 2 g/L DA solution. The treated PTFE film was immersed in DA solution and stirred continuously for a certain period of time (12 h, 24 h, 36 h, and 48 h) in an aerobic environment at room temperature. Ultrasonic radiation was used to remove the redundant PDA on the surface of the film to obtain a DA modified PTFE film (PTFE@PDA). Activated PTFE@PDA film was obtained by immersing in 0.5 mol/L AgNO_3_ solution for 1 h.

### 2.3. Preparation of PTFE@Ag

A total of 0.075 mol/L EDTA and ammonia was added into 0.5 mol/L AgNO_3_ solution until the mixed liquid became clear, and the pH was adjusted to 11 to 12 with 3 mol/L NaOH. A silver-plating solution was prepared by adding 0.1 mol/L glucose solution. The PTFE@PDA film was immersed vertically into the solution to ensure that the film contacted completely with the silver-plating solution. The film was ultrasonicated for 5 min with the ultrasound power of 200 W in order to fully moisten the sample surface and remove the bubbles. It was then washed with deionized water under ultrasonic radiation for 3 min, and dried in the air at room temperature. Finally, Ag plating film (PTFE@Ag) was obtained.

### 2.4. Preparation of Nickel Nanochains

Liquid-phase chemical reduction was employed to prepare nickel nanochains [36]. A total of 0.1 g NiCl_2_·6H_2_O was dissolved in a beaker containing 30 mL deionized water and 35 mL ethanol. An total of 1.3 g NaOH was added under stirring till it completely dissolved. A total of 6.0 mL N_2_H_4_·H_2_O solution was added, and it was subsequently water bath reacted in a 0.3 T magnetic field at 60 °C for 15 min. The black flocculent solids were nickel nanochains, which floated on mixed-solution surface. The prepared nickel nanochains were then separated with a magnet, and thoroughly washed with purified water and absolute ethanol, respectively. The nickel nanochains were finally obtained after vacuum drying at 60 °C for 2 h.

### 2.5. Preparation of PTFE@Ag+Ni@PVDF

A total of 1 g PVDF was mixed with 10 mL N-methylpyrrolidone and magnetically stirred for 3 h at 60 °C until the PVDF was completely dissolved. A total of 0.1 g nickel nanochains was added in the above solution, mixed, and stirred uniformly. The nickel nanochains and PVDF mixture were coated on a glass plate and vacuum dried at 60 °C for 5 h to obtain a PVDF casting film (Ni@PVDF). Three slices of PTFE@ Ag and two slices of Ni@PVDF film were piled one above another alternately with a “3+2” layered structure, and they were hot pressed for 5 min at 180 °C and 20 MPa. Finally, the PTFE@Ag+Ni@PVDF composite film was obtained (as shown in Figure 1).

### 2.6. Performance and Characterization of Samples

IR characterization was performed on a Brucker Vectex-70 infrared spectrometer, recording absorption spectra from wave number 4000 to 400 cm^−1^. XRD-6100 X-ray diffractometer with ST2253 tester was used to perform an XRD test, CuK α Radiation, λ = 0.15406 nm, scanning range 15°~80°, scanning speed 2°/min. The LakeShore vibrating sample magnetometer (7404, LakeShore Company, Kirkland, WA, USA) was used to test the magnetic induction strength. A Merlin-type, high-resolution field emission scanning electron microscope (Zeiss, Oberkochen, Germany) was used to view surface morphology of the thin film. The element distribution of Cu, Ni, and F was analyzed by using an energy spectrometer (EDS, Oxford, UK). The tensile performance was performed on a Metz CMT6104 electronic universal testing machine, with a tensile speed of 5.00 mm/min and a sample thickness of 100 μm at 25 °C. A bending test was performed as follows: the composite film was bent 180° forward and backward along the center of the measured point and maintained for 10 s, which was recorded as a single bending. Agilent PNA-N5244A Vector Network Analyzer (VNA) was used to measure electromagnetic shielding effectiveness. The test was based on GJB6190-2008 standard, with wave-guide mode selected for the sub item. The test frequency band was in the range of 8.2~12.4 GHz. The SE_T_ (total SE), SE_R_ (Reflection SE), SE_A_ (Absorption SE), and SE_M_ (Microwave multiple internal reflections SE) were obtained according to the S parameters (S11 and S21) [1,8,35].

## 3. Results and Discussions

### 3.1. Structure Analysis

The infrared spectra (IR) of microporous PTFE film and modified PTFE@PDA film are shown in Figure 2. It can be observed that the -OH stretching vibration peak appeared at 3367 cm^−1^ on the spectra of (a) to (d), and the absorption peak strength was stronger after modification due to the introduction of -OH by hydrophilic modification. The stretching vibration of the C-F bond in PTFE led to strong diffraction peaks at 1225 and 1155 cm^−1^. A diffraction peak at 1509 cm^−1^ exhibited in the PDA@PTFE film indicated the presence of C=C skeleton vibrations within the benzene ring and N-H vibration within the PDA. A bending vibration peak of N-H appeared at 1621 cm^−1^. It can be concluded that DA was polymerized on the surface of the microporous PTFE film. It could also be observed that the intensity of characteristic peaks of PDA from 1400 to 1650 cm^−1^ increased with the extension of surface modification time, indicating that the PDA content on the surface of PTFE increased. After modification of 24 h, the intensity of PDA peaks almost kept constant. Therefore, 24 h was chosen for the modification of PDA. Furthermore, the absorption peaks at 720 and 740 cm^−1^ were the amorphous structure in PTFE film, while the absorption peak from 510 to 640 cm^−1^ was the vibration of the C-F bond in PTFE. The peak at 625 cm^−1^ was the stretching vibrations of five adjacent carbons on the benzene ring of PDA. The peak intensity increased with time.

Figure 3 showed XRD patterns of PTFE, PTFE@PDA, PTFE@Ag, and PTFE@Ag+Ni/PVDF. There was a characteristic diffraction peak of PTFE film with high intensity at 2θ = 18.6°. After being modified by PDA, the peak was weakened in PTFE@PDA. This might be attributed to PDA adhesion and polymerization on the surface of PTFE, leading to the weakness of scatter ability to X-rays. The characteristic diffraction peaks appeared at 2θ = 37.6°, 43.8°, 63.9°, and 77.0°, corresponding to the (111), (200), (220), and (311) crystal faces of Ag, respectively. It was indicated that the PTFE film surface had been well covered by the Ag coating for PTFE@Ag film and PTFE@Ag+Ni/PVDF composite film; meanwhile, the intensity of diffraction peaks of PTFE@Ag+Ni/PVDF appearing at 2θ = 37.6° and 43.8° were enhanced, which was due to the densification of the silver layer caused by hot pressing. In addition, no AgO or Ag_2_O diffraction peaks were observed in the spectra, indicating the total deposition of the Ag element on PTFE film’s surface. Diffraction peaks of Ni could not be detected in the XRD spectra, which could be explained by the embedding of nickel nanochains in the film. This was consistent with the results of Cu composite films previously prepared by our research team [35].

### 3.2. Morphology Analysis

Figure 4 showed the SEM images with different magnification times and EDS photo of PTFE@Ag film. It can be observed from the figure that the surface of the PTFE film was relatively flat (Figure 4a). PTFE@Ag film presented a microporous structure with relatively complex changes in the three-dimensional structure of the pores after zooming (Figure 4b), such as mesh connectivity, pore curvature, and hole nest. It was possible that multiple micropores formed a single channel, and one micropore was connected to multiple channels. And the micropores on the PTFE film surface were almost filled. The Ag plating layer was dense, with a small grain size of about 100–200 nm. It can be observed that particle aggregation has occurred at some locations. It can be seen from the EDS photo of PTFE@Ag film (Figure 4c,d) that the entire film surface presented a uniform sky blue which represented the distribution of Ag elements, indicating that Ag was uniformly dispersed on the PTFE@Ag film’s surface. As a result, a layer of Ag was uniformly plated on PTFE film surface resulting in PTFE@Ag film.

Figure 5 was a cross-section SEM image (a) and EDS photo (b) as well as elemental mapping of Ag(c) and Ni(d)of PTFE@Ag+Ni@PVDF. To obtain cross-section SEM images of PTFE@Ag+Ni@PVDF film, the film was treated using liquid-nitrogen embrittlement fracturing. It can be observed in Figure 5a that the cross section was divided into a “3+2” sandwich-layered structure: the upper, middle, and lower three layers were of fine wire mesh structure PTFE@Ag, and the middle two layers of the film were nickel nanochains with a smooth and uniform surface. The total thickness of the film was about 80 μm. The PTFE@Ag film layer had obvious tearing and wire drawing phenomenon due to its strong toughness. Ag was plated on the surface of PTFE microporous film (as shown in Figure 5b), where the distribution of Ag elements was represented by blue dots. And it can be clearly observed in Figure 5c that there were three layers of Ag elements due to the three layers of PTFE@Ag film. The dispersion of nickel nanochains (Figure 5d) in the PVDF layer was assigned as green dots, indicating a relatively uniform distribution.

### 3.3. Performance Analysis

In the conductivity test, all parameters of the samples were inputted according to the actual measurement dimensions and UI testing was conducted. The UI testing reached 0.9999 or above before testing. The data used were the average value of the data from the three tests. The test results of electrical conductivity of PTFE@Ag film and PTFE@Ag+Ni@PVDF composite films are shown in Figure 6. The electrical conductivity of PTFE@Ag film was 5178.7 S·cm^−1^, and this value reached to 7507.5 S·cm^−1^ upon the preparation of the PTFE@Ag+Ni@PVDF composite film. It was maintained at 975.6 S·cm^−1^ after a 2000-cycle bending. The results were much higher compared to the PTFE@Cu+Ni@PVDF composite film [35]. It can be concluded that the composite film had excellent electrical conductivity, which might result in high-efficient electromagnetic interference shielding.

The magnetic properties of nickel nanochains, PTFE@Ag film, and PTFE@Ag+Ni@PVDF composite film were tested via using a vibrating sample magnetometer. Figure 7 showed the hysteresis loop of nickel nanochains, PTFE@Ag film, and the PTFE@Ag+Ni@PVDF composite film. There was almost a straight line for PTFE@Ag film, indicating that it nearly had no magnetic properties. An S-type hysteresis loop was observed for nickel nanochains and PTFE@Ag+Ni@PVDF composite film, indicating that they had a certain degree of magnetism. The saturation magnetization (*M_s_*) and the coercive force (*H_c_*) was 45.8 emu/g and 172 Oe, 35.8 emu/g and 103.2 Oe, respectively. According to the formula *M_s_* = ϕ*m_s_* (where ϕ is the volume fraction of particles and *m_s_* is the saturated magnetic moment of a single particle), it can be observed that the *M_s_* of PTFE@Ag+Ni@PVDF composite film mainly depended on the volume fraction of nickel nanochains [35]. Therefore, the magnetic loss caused by the addition of nickel nanochains in PTFE@Ag+Ni@PVDF composite film played a crucial role in enhancing the ability of the composite film to absorb electromagnetic waves. Furthermore, recent research showed that the construction of multilayer composite through structure control could effectively reduce the reflected EM waves and secondary EM radiation [37]. Nowadays, this is an effective method to reduce the high reflectivity via the layer-by-layer consumption process of EM waves. This process reduced secondary reflection and improved the absorption performance of the composite film. Therefore, a multilayered structure formed by alternating layers of PTFE@Ag and Ni@PVDF could generate multiple wave reflections. The magnetic loss caused by nickel nanochains made PTFE@Ag+Ni@PVDF composite film exhibit excellent electromagnetic shielding effectiveness.

Figure 8 was the electromagnetic shielding effectiveness diagram of PTFE@Ag film, PTFE@Ag+Ni@PDVF composite film, and PTFE@Ag+Ni@PDVF composite film after a 2000-cycle bending. The average electromagnetic shielding effectiveness of PTFE@Ag was 56.25 dB within the scope of 8.2~12.4 GHz electromagnetic wave frequency (X-band). The electromagnetic SE of PTFE@Ag+Ni@PDVF composite film was increased to 69.03 dB, and that of PTFE@Ag was increased by 22.72%, while electromagnetic SE remained at the value of 51.90 dB after a 2000-cycle bending. From the results, it can be concluded that PTFE@Ag+Ni@PDVF composite film was an electromagnetic-shielding film material with good electrical conductivity and medium electromagnetic SE. Besides the multilayered structure of the composite film, the internal porous structure also enhanced the local absorption and multiple reflections of the material. Abundant charge carriers were provided by Ag-coated PTFE polymer nanofibers, which were transferred freely through the Ag plating layer. Thus, interfacial polarization between the conductive Ag coating and the insulating polymer frame was enhanced. Incident electromagnetic waves were thereby effectively absorbed because of large charge storage ability of the polarized interface [38]. The rich porous structure of PTFE@Ag+Ni@PVDF could significantly increase the frequency of multiple reflections and scattering leading to effective improvement on shielding performance. Different electrical properties between nickel nanochains and PTFE polymer fibers could induce multiple internal reflection [39]. In addition, high electrical conductivity of nickel nanochains generated lattice vibrations as well as particle vibrations as a whole, leading to a reduced electronic free path compared to other conductors. Therefore, their magnetic domains were smaller than strong magnetic materials. This also generated a large hysteresis loss for electromagnetic waves; thus, the film exhibited good SE in both high- and low-frequency magnetic fields.

In addition, PTFE@Ag+Ni@PVDF composite film also had excellent tensile strength (~62.1 MPa) and strain (~134.4%) as shown in Figure 9. After a 2000-cycle bending, the value was still considerable. These novel properties make the composite film highly suitable for various applications in flexible devices.

We also made a comparison of the prepared PTFE@Ag+Ni@PVDF composite film to some reported lightweight electromagnetic-shielding materials, which are shown in Table 1. In our work, it was clearly demonstrated that the prepared composite film with a “3+2” structure exhibited high SE and conductivity.

## 4. Conclusions

A flexible, ultra-thin composite film with a “3+2” layered structure was successfully prepared by hot pressing of PTFE@Ag microporous film and Ni@PVDF casting film. This composite film was used as electromagnetic shielding. The thickness of the film was about 80 μm. It had a shielding effectiveness as high as 69.03 dB in the X-band range. This value could be maintained at 51.90 dB after a 2000-cycle bending. The electrical conductivity and saturation magnetization values of PTFE@Ag+Ni@PVDF composite film reached 7507.5 S/cm and 69.03 emu/g, respectively. In addition, the film also had excellent tensile strength (~62.1 MPa) and strain (~134.4%), which could meet the usage requirements in harsh environments. As a result, the prepared PTFE@Ag+Ni@PVDF composite film has great potential for applications in flexible devices and intelligent devices. These results point to the need for further research in designing new flexible electromagnetic shielding composite films.

## Figures and Tables

**Figure 1 materials-16-04831-f001:**
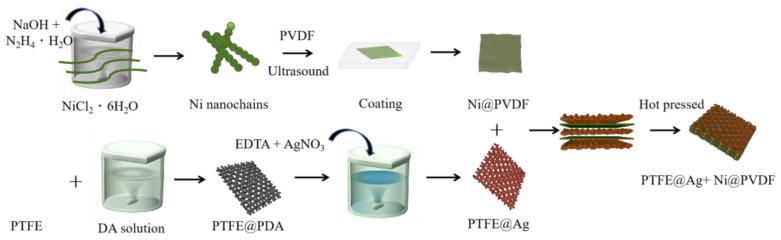
Schematic diagram for preparation of PTFE@Ag+Ni@PVDF composite film.

**Figure 2 materials-16-04831-f002:**
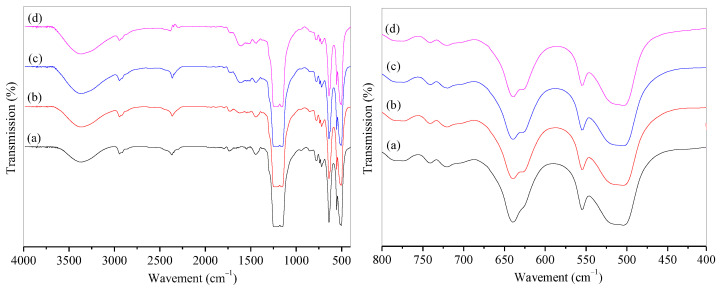
IR spectra of PTFE and PTFE@PDA films with modification of time: (a) PTFE; PTFE@PDA modified for (b) 12 h, (c) 24 h, and (d) 48 h.

**Figure 3 materials-16-04831-f003:**
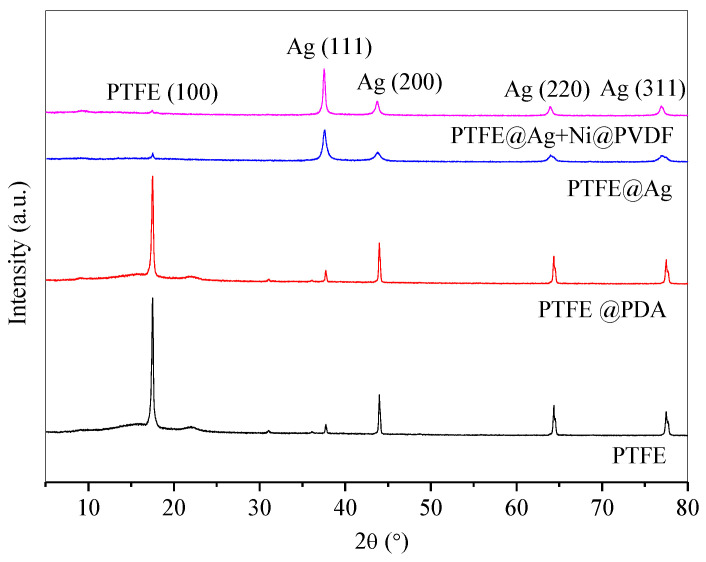
XRD patterns of PTFE, PTFE@PDA, PTFE@Ag, and PTFE@Ag+Ni@PVDF composite film.

**Figure 4 materials-16-04831-f004:**
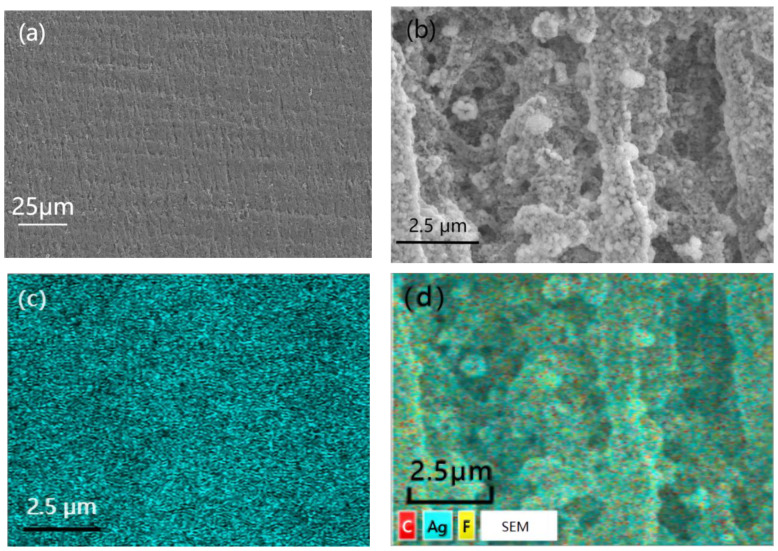
SEM images with different magnification times: (**a**) 500, (**b**) 2000, and EDS photo (**c**,**d**) of PTFE@Ag film.

**Figure 5 materials-16-04831-f005:**
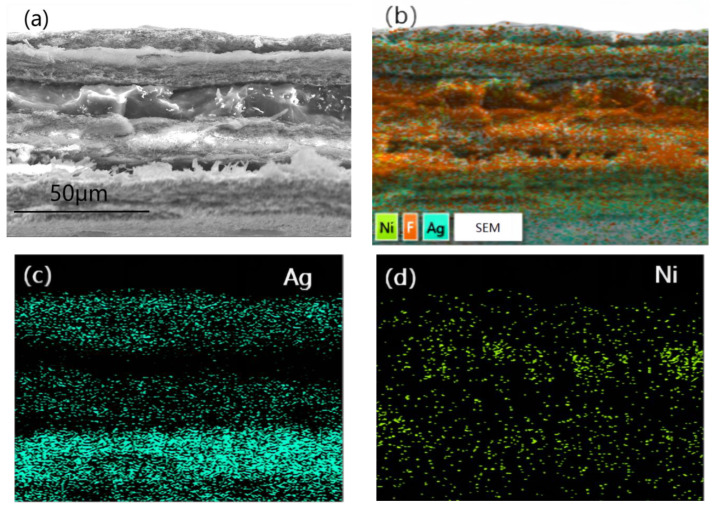
Cross-section SEM image (**a**) and EDS photo (**b**) as well as elemental mapping of Ag (**c**) and Ni (**d**) of PTFE@Ag+Ni@PVDF.

**Figure 6 materials-16-04831-f006:**
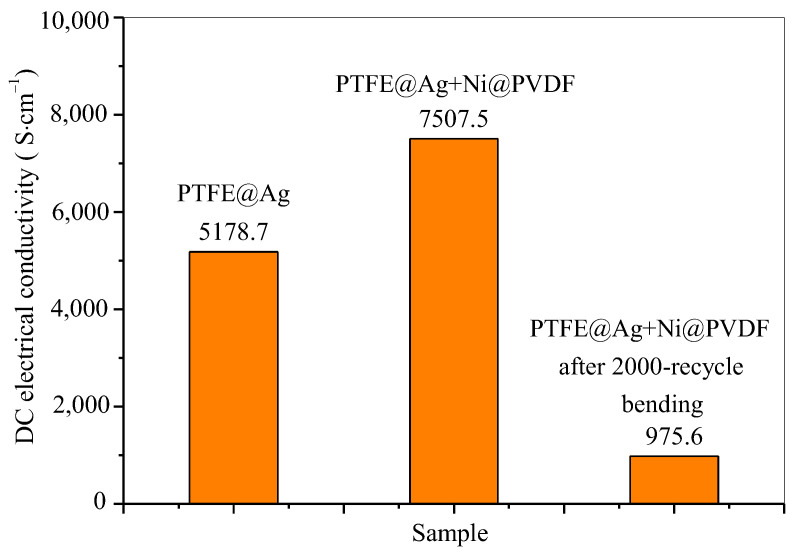
DC electrical conductivity of PTFE@Ag film and PTFE@Ag+Ni@PVDF composite film.

**Figure 7 materials-16-04831-f007:**
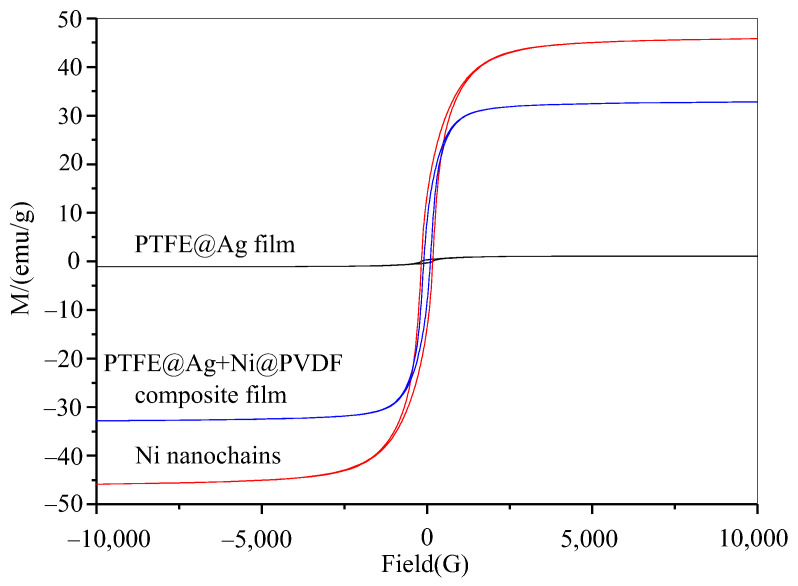
Hysteresis loop of Ni nanochains, PTFE@Ag film, and PTFE@Ag+Ni@PVDF composite film.

**Figure 8 materials-16-04831-f008:**
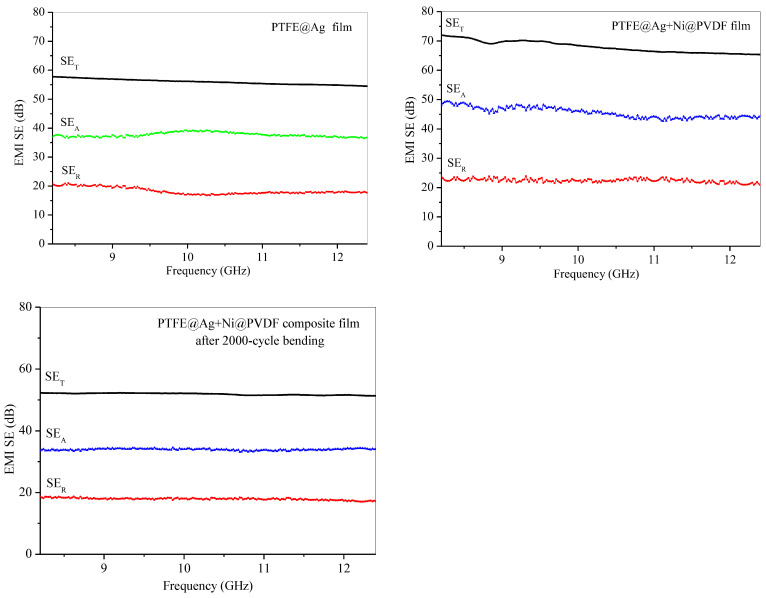
Electromagnetic shielding effectiveness diagram of PTFE@Ag film, PTFE@Ag+Ni@PDVF composite film, and PTFE@Ag+Ni@PDVF composite film after 2000-cycle bending.

**Figure 9 materials-16-04831-f009:**
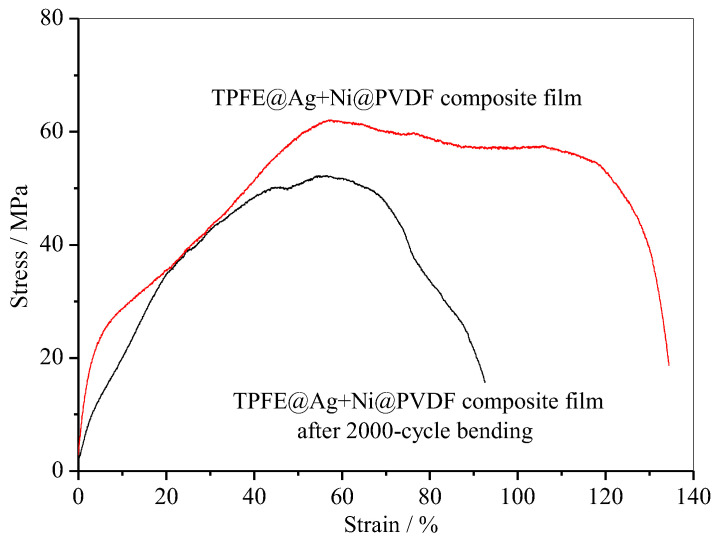
Tensile curve of PTFE@Ag+Ni@PVDF composite film.

**Table 1 materials-16-04831-t001:** Comparation of the EMI shielding performance for some reported lightweight electromagnetic-shielding materials.

Material	Conductivity (S/cm)	Thickness (mm)	EMI SE (dB)	Ref.
Ga-MXene/cellulose film	390.0	0.060	52.80	[1]
NWPU/PPy/AgNWs coating films	1026.1	0.480	64.20	[3]
EP/rGO/Ni-chainsx foams	0.151	—	41.11	[4]
MXene/jute composite	1519.0	3.000	59.08	[8]
PVDF-CNT-Graphene-NiCo chains composite films	912.0	3.000	63.30	[14]
EP/f-RGO/Ni-chains microcellular foam	~10^−1^	—	40.82	[17]
MXene/MrG composite membranes	2281.0	0.018	89.10	[19]
MXene/Galinstan composite films	4955.0	~0.015	66.28	[20]
graphene/AgNPs-coated CAFP	4431.0	0.035	92.29	[23]
EP/AgNW/TAGA	595.7	4.000	84.01	[24]
P@Ni-Co hybrid membrane	1139.6	0.180	77.80	[26]
PDA-Doped rGO/MWCNT composite	—	0.082	47.60	[33]
PTFE@Cu/Ni@PVDF composite films	1117.6	0.080	57.16	[35]
PTFE@Ag+Ni@PVDF composite films	7507.5	0.080	69.03	This work

## Data Availability

All relevant data are contained in the present manuscript.

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
