# Peer review of "Light Weight, Flexible and Ultrathin PTFE@Ag and Ni@PVDF Composite Film for High-Efficient Electromagnetic Interference Shielding"

_materials, 2023, doi:10.3390/ma16134831_

Round 1

Reviewer 1 Report

The article definitely contains some new and interesting results that can be recommended for publication, but only after the article has been finalized and some shortcomings have been eliminated.

1. Introduction needs more information about shielding.

2. Line 43-46. Note that there are other conductive polymers that have been successfully used recently for other optoelectronic applications:

V. P. Savchyn; et al. Low Temperature Physics 42, 597–600 (2016) https://doi.org/10.1063/1.4959020

Aksimentyeva, O. I., et al. "Modification of polymer-magnetic nanoparticles by luminescent and conducting substances." Molecular Crystals and Liquid Crystals 590 (2014): 35-42.

Kuznetsova, Lyubov S., et al. "Development of Nanocomposite Materials Based on Conductive Polymers for Using in Glucose Biosensor." Polymers 14 (2022): 1543

3. Line 47-59. This paragraph needs a few supporting references.

4. Lines 60-63. This short paragraph also needs supporting references.

5. Figure 2. Describe in detail the part of the spectrum in the region of 400-800 cm-1 and the nature of its evolution.

6. Line 179. If Ni could not be detected in the XRD spectra, how else can one see it?

7. Figures 4 and 5 contain the legend in Chinese.

8. Figure 6. This seems to be too precise. Can you show provide error bars and corresponding  explanations in the text?

9. The authors described all the experiments in sufficient detail, however, in the conclusions it should be clearly stated what fundamentally new data on the studied materials were obtained in this work.

Author Response

Dear Iulia Ilies and reviewers,

Thank you very much for giving us the chance to revise our manuscript (“Light weight, Flexible and ultrathin PTFE@Ag and Ni@PVDF composite film for high-efficient electromagnetic interference shielding”, ID: materials-2469514). Based on the editor’s and reviewers’ comments, we have revised our manuscript carefully. Details are as follows:

Reviewers' comments:

Reviewer #1: 

  1. Introduction needs more information about shielding.

Reply:

Thank you very much for your comment. We have reorganized and polished the introduction and added related literature in the references to highlight the research background of shielding as follows as well as in the revised manuscript.

At present, polyacrylonitrile (PAN)[9], polyvinylalcohol (PVA)[11], PTFE[12,13], PVDF[14,15], Polyethylene terephthalate (PET)[16] etc. are used as polymers matrix, and the commonly used conductive fillers mainly include graphene [4,17-19], MXenes [8,19-23], metals[16,24,25], alloys [14,26], perovskite metal oxides [27,28] and so forth. These conductive polymer composite films exhibited good EMI SE. For example, Tan H.X. et. al. designed and prepared a sandwich-structured Ti3C2Tx-MXene/ reduced-graphene-oxide composite (MrG) membrane with high conductivities and abundant interior blowholes for highperformance EMI shielding and IR stealth. The membrane had distinctive porous structures and was about 18-μm-thickness with a high EMI SE of 89.1 dB and a satisfactory absolute EMI SE (SSE/t) of up to 55282.8 dB cm2/g. This was the best result for MXene-based nanocomposites with similar thickness[19]. The distinctive porous structure was believed to play an initial role in improving the EMI SE.

  1. Kim,R.; Fujimori,K.; Kim, B.S.; et al. Lightweight nanofibrous EMI shielding nanowebs prepared by electrospinning and metallization. Compos. Sci. Technol. 2012, 72, 1233-1239.
  2. Sang,M.; Liu, G.H.;Liu, S.; et al. Flexible PTFE/MXene/PI soft electrothermal actuator with electromagnetic-interference shielding property. Engin. J. 2021, 414, 128883.
  3. Guo,B.Z.; Liang, J.Y.; Chen, J.F. etc. Highly flexible and ultrathin electromagneticinterference-shielding film with a sandwich structure based on PTFE@Cu and Ni@PVDF nanocomposite materials. RSC Adv.2022, 12, 29688-29696.
  4. Yang, R.; Zhou,Y.Y.;Ren, Y.M.; et al. Promising PVDF-CNT-Graphene-NiCo chains composite films with excellent electromagnetic interference shielding performance. J. Alloys Comp.2022, 908, 164538.
  5. Ye, L.X.; Liu,L.X.; Yin, G.; et al. Highly conductive, hydrophobic, and acid/alkali-resistant MXene@PVDF hollow core-shell fibers for efficient electromagnetic interference shielding and Joule heating. Mater. Today Phys.2023, 35, 101100.
  6. Kuznetsova, L.S.; Arlyapov, V. A.; Kamanina, O.A.; et al. Development of nanocomposite materials based on conductive polymers for using in glucose biosensor. Polymers2022, 14, 1543.
  7. Tan,H.X.; Gou, J.R.; Zhang, X.; et al. Sandwich-structured Ti3C2Tx-MXene/reduced-graphene-oxide composite membranes for high-performance electromagnetic interference and infrared shielding. J. Membrane Sci.2023, 675, 121560. 
  8. Tao, J.Q.;Liu,Ni.; Li, S.X.; et al. Structural manipulation of silver nanowire transparent conductive films for optoelectrical property optimization in different application fields. Thin Solid Films 2021, 729, 138679.
  9. Savchyn, V. P.; Popov, A. I.; Aksimentyeva,O.I.; et al. Cathodoluminescence characterization of polystyrene-BaZrO3hybrid composites. Low Temp. Phys.2016, 42, 597–600.
  10. Aksimentyeva, O. I.; Savchyn, V. P.; Dyakonov, V. P.; et al. Modification of polymer-magnetic nanoparticles by luminescent and conducting substances. Mol. Cryst. Liq. Crys.2014, 590, 35-42.

We also made a comparison of the prepared PTFE@Ag+Ni@PVDF composite film to some reported lightweight electromagnetic-shielding materials, which were shown in Table 1. It was clearly showed that the composite film with a “3+2” structure exhibiting high SE and conductivity in our work.

Table 1 Comparation of the EMI shielding performance for some reported lightweight electromagnetic-shielding materials

Material

Conductivity

[S/cm]

 Thickness

 [mm]

 EMI SE

 [dB]

Ref.

Ga- MXene/cellulose film

390.0

0.060

 52.80

1

NWPU/PPy/AgNWs coating films

1026.1

0.480

64.20

3

EP/rGO/Ni-chainsx foams

0.151

 41.11

4

MXene/jute composite

1519.0

3.000

 59.08

8

 PVDF-CNT-Graphene-NiCo chains composite films

912.0

3.000

63.30

14

EP/f-RGO/Ni-chains microcellular foam

 ~10− 1

40.82

17

MXene/MrG composite membranes

2281.0

0.018

89.10

19

 MXene/Galinstan composite films

4955.0

 ~0.015

66.28

20

graphene/AgNPs-coated CAFP

4431.0

0.035

 92.29

23

EP/AgNW/TAGA

595.7

4.000

84.01

24

P@Ni-Co hybrid membrane

1139.6

0.180

77.80

26

PDA-Doped rGO/MWCNT composite

0.082

47.60

33

PTFE@Cu/Ni@PVDF composite films

 1117.6

0.080

57.16

35

PTFE@Ag+Ni@PVDF composite films

7507.5

0.080

69.03

This work

  1. Line 43-46. Note that there are other conductive polymers that have been successfully used recently for other optoelectronic applications:
  2. P. Savchyn; et al. Low Temperature Physics 42, 597–600 (2016) https://doi.org/10.1063/1.4959020

Aksimentyeva, O. I., et al. "Modification of polymer-magnetic nanoparticles by luminescent and conducting substances." Molecular Crystals and Liquid Crystals 590 (2014): 35-42.

Kuznetsova, Lyubov S., et al. "Development of Nanocomposite Materials Based on Conductive Polymers for Using in Glucose Biosensor." Polymers 14 (2022): 1543

Reply:

Thank you very much for your comment. We have rewritten the sentence and introduced the above listed references into the sentence in the follows as well as in the revised manuscript.

At present, polyacrylonitrile (PAN)[9], polyvinylalcohol (PVA)[11], PTFE[12,13], PVDF[14,15], Polyethylene terephthalate (PET)[16] etc. are used as polymers matrix, and the commonly used conductive fillers mainly include graphene [4,17-19], MXenes [8,19-23], metals[16,24,25], alloys [14,26], perovskite metal oxides [27,28] and so forth. 

  1. Kuznetsova, L.S.; Arlyapov, V. A.; Kamanina, O.A.; et al. Development of nanocomposite materials based on conductive polymers for using in glucose biosensor. Polymers2022, 14, 1543.
  2. Savchyn, V. P.; Popov, A. I.; Aksimentyeva,O.I.; et al. Cathodoluminescence characterization of polystyrene-BaZrO3hybrid composites. Low Temp. Phys.2016, 42, 597–600.
  3. Aksimentyeva, O. I.; Savchyn, V. P.; Dyakonov, V. P.; et al. Modification of polymer-magnetic nanoparticles by luminescent and conducting substances. Mol. Cryst. Liq. Crys.2014, 590, 35-42.
  4. Line 47-59. This paragraph needs a few supporting references.

Reply:

Thank you very much for your comment. We have added related literature in the references as follows as well as in the revised manuscript.

Polytetrafluoroethylene (PTFE) has excellent characteristics such as high temperature resistance, corrosion resistance, low adhesion, flow resistance and friction coefficient[29,30]. PTFE electromagnetic shielding materials is believed to have great practical application potential[13]. However, they are prone to slippage between fillers and polymer substrates under large deformation conditions, resulting in damage to the conductive network[31]. Therefore, electromagnetic shielding effectiveness decreased, and the metal layer is easy to fall off, resulting in poor secondary processing. Studies have shown that dopamine would undergo an oxidation self polymerization reaction to form a strong polydopamine layer (PDA) attached to the surface of solid materials under weak alkaline solution conditions[30,32]. The phenolic hydroxyl, amino, and imino functional groups in PDA could change the surface properties of the coated material, providing a chelating site with metal ions, as well as favorable conditions for subsequent modification operations such as grafting, plating, and coating[33].

  1. Zhang,D.X.;Zhuo,L.;Xiang,Q. Electrophoretic deposition of polytetrafluoro -ethylene (PTFE) as anti-corrosion coatings. Lett.2023, 346, 134524.
  2. Yu,Y.F.; Zhang, L.; Li, X.D., et al. Multifunctionalization of PTFE membrane surface for biofouling resistance and oil/water separation performance improvement. Environ. Chem. Eng. 2023, 11, 109158.
  3. Das, A.; Hayvaci, H.T.; Tiwari, M.K.; et al. Superhydrophobic and conductive carbon nanofiber/PTFE composite coatings for EMI shielding. J. Colloid Interf. Sci.2011, 353, 311-315.
  4. Yang, Y.; Liu, Y.X.;Deng, B.W.; et al. Construction of three-dimensional carbon materials-based conductive bonding network in flexible supercapacitor electrodes. Electrochimica Acta 2023, 440, 141751.   
  5. Shahkaramipour, N.; Lai, C.K.; Venna, S.R.; et al. Membrane surface modification using thiol-containing zwitterionic polymers via bioadhesive polydopamine.Ind. Eng. Chem. Res.2018, 57, 2336-2345.
  6. Sharma,S.; Lee, J.; Dang, T.T.; et al. Ultrathin freestanding PDA-Doped rGO/ MWCNT composite paper for electromagnetic interference shielding applications. Chem. Eng. J. 2022, 430, 132808.
  7. Lines 60-63. This short paragraph also needs supporting references.

Reply:

Thank you very much for your comment. We have added related literature in the references as follows as well as in the revised manuscript.

As a filler material with excellent electrical conductivity, silver was widely used in conductive films[23-25,34]. However, due to the high toxicity of stannous chloride in traditional silver plating chemical reagents and the high price of palladium chloride, it is necessary to find more ideal alternatives in practical applications.

  1. Wang, P.C.; Jian, M.L.; WuM.J.Q.; et al. Highly sandwich-structured silver nanowire hybrid transparent conductive films for flexible transparent heater applications. Part A 2022, 159, 106998.
  2. Figure 2. Describe in detail the part of the spectrum in the region of 400-800 cm-1 and the nature of its evolution.

Reply:

Thank you very much for your comment. We have redraw a detailed spectrum and described the part of the spectrum in the region of 400-800 cm-1 and the nature of its evolution in detail. The revised part is shown in the follow as well as in the revised manuscript.

Furthermore, the absorpiton peak at 720 and 740 cm-1 was amorphous structure in PTFE film, while the absorpiton peak at 510 to 640 cm-1 was the vibration of the C-F bond in PTFE. The peak at 625 cm-1 was the stretching vibrations of five adjacent carbons on the benzene ring of PDA. The peak intensity was increased with time.

  1. Line 179. If Ni could not be detected in the XRD spectra, how else can one see it?

Reply:

Thank you very much for your comment. Ni could not be detected in the XRD spectra, but could be seen in SEM and EDS test as shown in Figure 5(d). The dispersion of nickel nanochains (Fig.5(d)) in the PVDF layer was assigned as green dots.

  1. Figures 4 and 5 contain the legend in Chinese.

Reply:

Thank you very much for your comment. The Chinese words has been replaced by English words in the photo as shown in the following Figure.

  1. Figure 6. This seems to be too precise. Can you show provide error bars and corresponding explanations in the text?

Reply:

Thank you very much for your comment. We are not sure if our understanding is correct. All parameters of the samples were inputted according to the actual measurement dimensions and UI testing was conducted. The UI testing reached 0.9999 or above before testing. A screenshot of the original data is as follows.

All the data used to draw Figure 6 were the average value of the three test data. Therefore, only one precise data was used in Figure 6. We have explained the source of the data in the text as follows as well as revised manuscript.

In the conductivity test, all parameters of the samples were inputted according to the actual measurement dimensions and UI testing was conducted. The UI testing reached 0.9999 or above before testing. The data used were the average value of the three test data.

  1. The authors described all the experiments in sufficient detail, however, in the conclusions it should be clearly stated what fundamentally new data on the studied materials were obtained in this work.

Reply:

Thank you very much for your comment. We have rewritten the conclusion section  to make a clear statement of what fundamentally new data on the studied materials were obtained in this work as follows as well as in the revised manuscript.

A flexible, ultra thin composite film with “3+2” layered structure were successfully prepared by hot press of PTFE@Ag microporous film and Ni@PVDF casting film. This composite film was used as electromagnetic shielding. The thickness of the film was about 80 μm and it played a shielding effectiveness as high as 69.03 dB in the X-band range. This value could be maintained at 51.90 dB after 2000-cycle bending. The electrical conductivity and saturation magnetization values of PTFE@Ag+Ni@PVDF composite film reached 7507.5 S/cm and 69.03 emu/g, respectively. In addition, the film also had excellent tensile strength (~62.1 MPa) and strain (~134.4%), which could meet the usage requirements in harsh environments. As a result, the prepared PTFE@Ag+Ni@PVDF composite film had great potentials in applications of flexible devices and intelligent devices. It also pointed the research direction for designing new flexible electromagnetic shielding composite films.

Reviewer 2 Report

Please see the enclosed comment file.

Need moderate changes.

Author Response

Dear Iulia Ilies and reviewers,

Thank you very much for giving us the chance to revise our manuscript (“Light weight, Flexible and ultrathin PTFE@Ag and Ni@PVDF composite film for high-efficient electromagnetic interference shielding”, ID: materials-2469514). Based on the editor’s and reviewers’ comments, we have revised our manuscript carefully. Details are as follows:

Reviewers' comments:

Reviewer #2: 

  1. Abstract: Define the abbreviation PDA before use.

Reply:

Thank you very much for your comment. We have defined PDA in Abstract as follows as well as in the revised manuscript.

Dopamine was used to modify polytetrafluoroethylene (PTFE) in order to obtain functional polydopamine (PDA) surface-modified PTFE microporous film (PTFE@PDA).

  1. Authors wrote:

“Long-time explosion to electromagnetic wave…”

Replace “explosion” with “exposure”.

Reply:

Thank you very much for your comment. We have replaced “explosion” with “exposure” as follows as well as in the revised manuscript.

Long-time exposure to electromagnetic wave may lead to serious harm of human health, and electromagnetic wave leakage can also endanger information security [1].

  1. Authors wrote:

“Conductivity of a material is generally considered to be one of the most critical parameters affecting SE, which is related to the material surface structure, conductivity and magnetism[5].”

Saying conductivity of a material is dependent on the conductivity is meaningless. I believe authors are referring to the “intrinsic conductivity” of material. Please rephrase the sentence.

Reply:

Thank you very much for your comment. We have rephrased the sentence as follows as well as in the revised manuscript.

Intrinsic conductivity of a material is generally considered to be one of the most critical parameters affecting SE, which is related to the material surface structure and thickness [5]. 

  1. Rephrase

“Nowadays, commonly used conductive fillers consisting of graphene [4,11,12], MXenes [8,13-16], metals and alloys [17-19], and so forth.”

Reply:

Thank you very much for your comment. We have rephrased the sentence as follows as well as in the revised manuscript.

At present, polyacrylonitrile (PAN)[9], polyvinylalcohol (PVA)[11], PTFE[12,13], PVDF[14,15], Polyethylene terephthalate (PET)[16] etc. are used as polymers matrix, and the commonly used conductive fillers mainly include graphene [4,17-19], MXenes [8,19-23], metals[16,24,25], alloys [14,26], perovskite metal oxides [27,28] and so forth.

  1. Kim,R.; Fujimori,K.; Kim, B.S.; et al. Lightweight nanofibrous EMI shielding nanowebs prepared by electrospinning and metallization. Compos. Sci. Technol. 2012, 72, 1233-1239.
  2. Sang,M.; Liu, G.H.;Liu, S.; et al. Flexible PTFE/MXene/PI soft electrothermal actuator with electromagnetic-interference shielding property. Engin. J. 2021, 414, 128883.
  3. Guo,B.Z.; Liang, J.Y.; Chen, J.F. etc. Highly flexible and ultrathin electromagneticinterference-shielding film with a sandwich structure based on PTFE@Cu and Ni@PVDF nanocomposite materials. RSC Adv.2022, 12, 29688-29696.
  4. Yang, R.; Zhou,Y.Y.;Ren, Y.M.; et al. Promising PVDF-CNT-Graphene-NiCo chains composite films with excellent electromagnetic interference shielding performance. J. Alloys Comp.2022, 908, 164538.
  5. Ye, L.X.; Liu,L.X.; Yin, G.; et al. Highly conductive, hydrophobic, and acid/alkali-resistant MXene@PVDF hollow core-shell fibers for efficient electromagnetic interference shielding and Joule heating. Mater. Today Phys.2023, 35, 101100.
  6. Kuznetsova, L.S.; Arlyapov, V. A.; Kamanina, O.A.; et al. Development of nanocomposite materials based on conductive polymers for using in glucose biosensor. Polymers2022, 14, 1543.
  7. Tan,H.X.; Gou, J.R.; Zhang, X.; et al. Sandwich-structured Ti3C2Tx-MXene/reduced-graphene-oxide composite membranes for high-performance electromagnetic interference and infrared shielding. J. Membrane Sci.2023, 675, 121560. 
  8. Tao, J.Q.;Liu,Ni.; Li, S.X.; et al. Structural manipulation of silver nanowire transparent conductive films for optoelectrical property optimization in different application fields. Thin Solid Films 2021, 729, 138679.
  9. Savchyn, V. P.; Popov, A. I.; Aksimentyeva,O.I.; et al. Cathodoluminescence characterization of polystyrene-BaZrO3hybrid composites. Low Temp. Phys.2016, 42, 597–600.
  10. Aksimentyeva, O. I.; Savchyn, V. P.; Dyakonov, V. P.; et al. Modification of polymer-magnetic nanoparticles by luminescent and conducting substances. Mol. Cryst. Liq. Crys.2014, 590, 35-42.
  11. Authors wrote:

“However, they are prone to slippage between fillers and polymer substrates under large deformation conditions, resulting in damage to the conductive network.”

If authors are mentioning polymer/filler composite, saying “substrate” is not appropriate. Instead, it should be “matrix” (i.e. polymer matrix/fillers).

Reply:

Thank you very much for your comment. We have replaced “substrate” with “matrix” in the sentence as follows as well as in the revised manuscript.

However, they are prone to slippage between fillers and polymer matrix under large deformation conditions, resulting in damage to the conductive network[25].

  1. Fix typos in NiCl2 6H2O, 45℃, AgNO3, NiCl2·6H2O, N2H4·H2O. Check and fix such errors throughout the manuscript.

Reply:

Thank you very much for your comment. We have checked and fixed such errors throughout the manuscript. It was shown as follows as well as in the revised manuscript in detail.

AgNO3,  NiCl2 · 6H2O,  N2H4 · H2O,  cm-1

  1. Figure 5.

To obtain cross-section SEM images of the films how the samples were prepared, please explain.

Reply:

Thank you very much for your comment. We have explained how the samples were prepared in the manuscript as follows as well as in the revised manuscript.

To obtain cross-section SEM images of PTFE@Ag + Ni@PVDF film, the film was  treated by a liquid-nitrogen embrittlement fracture. 

  1. Authors wrote:

“And It can be clearly observed in Fig.5(c) that three were three layers of Ag elements due to the three layer of PTFE@Ag film.”

What is “three were three layers”?

Reply:

Thank you very much for your comment. The word “there” was mistakenly written as “three”. So the correct sentence is “ And It could be clearly observed in Fig.5(c) that there were three layers of Ag elements due to the three layer of PTFE@Ag film.” 

This was also corrected in the revised manuscript.

  1. What was the thickness of PTFE@Ag + Ni@PVDF composite film? Please mention it in the experimental section.

Reply:

Thank you very much for your comment. We have mentioned the thickness of PTFE@Ag + Ni@PVDF composite film as follows as well as in the revised manuscript.

The thickness of the film was about 80 μm.

  1. Authors wrote:

“According to formula M=φMs, it can be observed that the Ms of PTFE@Ag+Ni@PVDF composite film was mainly depended on the volume fraction of nickel nanochains[20].”

What is φ? Please keep all the variables in slanted font.

Reply:

Thank you very much for your comment. We have defined Ï• in the formula Ms=Ï•ms, and revised all the variables in slanted font as follows as well as in the revised manuscript.

According to formula Ms=Ï•ms (where Ï• is the volume fraction of particles and ms is the saturated magnetic moment of a single particle), it can be observed that the Ms of PTFE@Ag+Ni@PVDF composite film was mainly depended on the volume fraction of nickel nanochains[35].

  1. Two different notations in “. The SET(total SE), SER (Reflection SE), SEA (Absorption SE),” and in Figure 8. Keep consistency.

Reply:

Thank you very much for your comment. We have revised the notations in text as follows as well as in the revised manuscript. Therefore, the notations in text are consistent with figure 8.

The SET (total SE), SER (Reflection SE), SEA (Absorption SE), and SEM (Microwave multiple internal reflections SE) were obtained according to the S parameters (S11 and S21) [1,8,35].

  1. Authors wrote:

“After 2000-cycle bending, this value still remained at 51.90 dB.”

There is no description of how the bending cycle tests were performed. What was the curvature of the bending cycle? Please explain briefly in the experimental section.

Reply:

Thank you very much for your comment. We have described how the bending cycle tests were performed as follows as well as in the revised manuscript part 2.6.

A bending test was performed as follows: the composite film was bent 180° forward and backward along the center of the measured point and maintained for 10 s, which was recorded as a single bending.

  1. There is no mention of the adhesion strength between PTFE@Ag film and Ni@PVDF film.

Reply:

Thank you very much for your comment. The PTFE@Ag and Ni@PVDF film were prepared by hot pressed at 180℃ and 20MPa for 5 minutes to get the PTFE@Ag+Ni@PVDF composite film. The adhesion between them was tested by repeatedly bending the composite film. After 2000-recycle bending, the composite film did not crack and maintained good electromagnetic shielding performance.

  1. Figure 1 appears to have a problem. I noticed that the word “hot-pressed” appears twice, which is not in consistent with the experimental description.

Reply:

Thank you very much for your comment. We have redrawn Figure 1 as follows as well as in the revised manuscript.

  1. Finally, the authors need to make an effort to go through line by line to find the grammatical errors.

Reply:

Thank you very much for your comment. We have gone through the entire manuscript and modified the English writing. Details could be seen in the revised manuscript.

Round 2

Reviewer 1 Report

The authors have successfully improved the original version of their manuscript, responding constructively to all the comments/recommendations of the reviewer.  Therefore, the article can be recommended for publication.

Reviewer 2 Report

The changes are acceptable and therefore, I recommend accepting the article for publication.